**Assessing global-scale organic matter reactivity patterns in marine**
**sediments using a lognormal reactive continuum model**
**Sinan Xu[1,2,3], Bo Liu[3], Sandra Arndt[4], Sabine Kasten[3,5], and Zijun Wu[1*]**
[1]State Key Laboratory of Marine Geology, School of Ocean and Earth Science, Tongji
University, Shanghai, 200092, China
[2]The Key Laboratory of Gas Hydrate, Ministry of Natural Resources, Qingdao Institute of
Marine Geology, Qingdao, 266071, China
[3]Alfred Wegener Institute Helmholtz Centre for Polar and Marine Research, 27570
Bremerhaven, Germany
[4]Department of Geosciences, Environment and Society, Universit´e Libre de Bruxelles,
Avenue Franklin Roosevelt 50, 1050 Brussels, Belgium
[5]Faculty of Geosciences, University of Bremen, 28359 Bremen, Germany
**Correspondence to: Zijun Wu (wuzj@tongji.edu.cn)**

## Abstract

Organic matter (OM) degradation in marine sediments is largely controlled by its
reactivity and profoundly affects the global carbon cycle. Yet, there is currently no general
framework that can constrain OM reactivity on a global scale. In this study, we propose a
reactive continuum model based on a lognormal distribution ($l$-RCM), where OM
reactivity is fully described by parameters $\mu$ (the mean reactivity of the initial OM bulk
mixture) and $\sigma$ (the variance of OM components around the mean reactivity). We use the
$l$-RCM to inversely determine $\mu$ and $\sigma$ at 123 sites across the global ocean. The results show
that the apparent OM reactivity ($<k>=\mu\cdot\exp(\sigma^2/2)$) decreases with decreasing
sedimentation rate ($\omega$) and show that OM reactivity is more than three orders of magnitude
higher in shelf than that in abyssal regions. Despite the general global trends, higher than
expected OM reactivity is observed in certain ocean regions characterized by great water
depth and/or pronounced oxygen minimum zones, such as the Eastern-Western Coastal
Equatorial Pacific and the Arabian Sea, emphasizing the complex control of the
depositional environment (e.g., OM flux, oxygen content in the water column) on benthic
OM reactivity. Notably, the *l*-RCM can also highlight the variability of OM reactivity in
these regions. Based on inverse modeling results in our dataset, we establish the significant
statistical relationships between $<k>$ and $\omega$, and further map the global OM reactivity
distribution. The novelty of this study lies in its unifying view, but also in contributing a
new framework that allows predicting OM reactivity in data-poor areas based on readily
available (or more easily obtainable) information. Such a framework is currently lacking
and limits our abilities to constrain OM reactivity in global biogeochemical and/or Earth
System Models.

**1 Introduction**
Marine sediments act as the ultimate sink for organic carbon. The size and reactivity of
the benthic organic matter (OM) reservoir is a critical component of the global carbon cycle
(Arndt et al., 2013). In particular, the reactivity of benthic OM imposes a substantial control
on the magnitude of benthic carbon export and burial over geological timescales due to the
recycling of organic carbon by dissimilatory microbial activity in the deep biosphere
(Boudreau, 1992; Zonneveld et al., 2010), the dissolution and precipitation of carbonates
(Meister et al., 2022; Nöthen and Kasten, 2011), and the production of methane (Dickens
et al., 2004; Whiticar, 1999). Decades of research have shown that OM reactivity is
controlled by both the nature of the OM (origin, composition and degradation state), as
well as its environmental and depositional conditions (e.g., redox conditions, sedimentation
rate, mineral protection, microbial community composition and biological mixing)
(Burdige, 2007; Egger et al., 2018; Hartnett et al., 1998; Hedges and Keil, 1995; Larowe
et al., 2020a; Zonneveld et al., 2010). However, due to the complex and dynamic nature of
the main controls on OM reactivity, the specific relative significance of these controlling
factors remains poorly quantified. Consequently, OM degradation models generally do not
explicitly describe the influence of environmental and depositional factors on OM
reactivity and its evolution but rather apply simplified parametrizations (Freitas et al., 2021;
Pika et al., 2021). Over the past decades, several models have been developed and
successfully used to quantify OM degradation in marine sediments. They can be broadly
divided into two groups: discrete models, such as the (multi) $G$ model (Berner, 1964;
Jørgensen, 1978), and continuum models, such as the reactive continuum model (RCM)
(Boudreau and Ruddick, 1991) and the power model (Middelburg, 1989).
Discrete models divide the bulk OM pool into several discrete fractions, each with its
own constant reactivity (Fig.1A). The 1-$G$ model is the earliest OM degradation model,
which is based on the assumption that OM degrades according to first order dynamics with
a single constant degradation rate constant (Berner, 1964). The multi-$G$ model, on the other
hand, divides OM into several fractions, and each fraction is degraded according to a first-
order rate with a fraction-specific reactivity (Jørgensen, 1978). Although multi-$G$ models
successfully fit observed OM degradation dynamics when comprehensive data sets are
available, their application on a global scale is complicated by the need to partition the OM
reactivity into a finite number of fractions and define their reactivities. A multi-$G$ model
with $n$ discrete OM fractions requires constraining $2n$-1 parameters and is, thus, over-
parametrized (Jørgensen, 1978). Nevertheless, because of its mathematical simplicity and
wide use, multi-$G$ models have been used in a range of diagenetic models designed for the
global/regional scale (e.g., CANDI, MEDIA, MEDUSA, and OMEN SED) (Boudreau,
1996; Meysman et al., 2003; Munhoven, 2007; Pika et al., 2021). Constraining the $2n$-1
OM  degradation  model  parameters  for  these  global-scale  applications  is  not
straightforward. Early strategies for constraining the reactivity of OM on a global scale
have focused on deriving empirical relationships between OM reactivity and single, easily
observable characteristics of the depositional environment (water depth, sedimentation rate,
or OM flux) (Arndt et al., 2013). However, poor statistically significant link between OM
reactivity and depositional environment could be established ($R^2$<0.1) after compiling
published multi-$G$ model's parameters across a wide range of depositional environments,
model complexities as well sediment depths/ burial time scales (Arndt et al., 2013).
Reactive continuum models (RCMs) are an alternative to discrete models. They assume
that OM compounds are continuously distributed over a wide range of reactivities. The
degradation rate can be described as the sum of an infinite number of discrete fractions,
each degraded according to first-order kinetics (Boudreau and Ruddick, 1991), as
$$G(t) = \int_0^\infty G(0) \cdot g(k,0) \cdot e^{-kt} dk \qquad (1)$$
where $G(t)$ is OM content at time $t$, $G(0)$ is OM content at the sediment-water interface
(SWI), $k$ is the first-order degradation rate constant, and $g(k,0)$ is the initial reactivity
distribution of OM at the SWI. The key to constructing an RCM is to select a continuum
distribution that describes the OM reactivity at the SWI (Fig. 1B). Considering the $k$ value
in Eq. 1 must be greater than zero ($k > 0$), some of the all-axial statistical distributions ($x \in$
($-\infty$, $+\infty$)) are not appropriate for constructing RCM (e.g., Normal distribution, Fig.1D$_1$).
Boudreau and Ruddick. (1991), following Aris (1968) and Ho et al. (1987), proposed to
use a Gamma distribution ($\gamma$-RCM, Fig.1D$_2$) due to its mathematical properties and its
ability to capture the observed dynamics:
$$g(k, 0) = \frac{a^v \cdot k^{v-1} \cdot e^{-ak}}{\Gamma(v)} \tag{2}$$

where $a$ is the average age of the OM at the SWI, $v$ is the shape parameter, and $\Gamma(v)$ is the
Gamma function. In addition, Middelburg. (1989) empirically derived a power law from a
large data compilation of measured OM reactivity (Fig.1C), which is mathematically
equivalent to the $\gamma$-RCM. The advantage of the continuum models over the discrete models
is that they merely require constraining two free parameters to capture the widely observed
continuous decrease in OM reactivity with degradation time/depth. Recently, $\gamma$-RCM has
been used to inversely determine the free $\gamma$-RCM parameters, and thus benthic OM
reactivity, from observed POC and sulfate depth profiles across a wide range of different
depositional environments (Freitas et al., 2021). Although results revealed broad global
patterns, no significant statistical relationship ($R^2 < 0.46$) between the parameters ($a$ and $v$)
of the $\gamma$-RCM (Arndt et al., 2013) and characteristics of the depositional environment could
be found, and constraining OM degradation model parameters on the global scale thus
remains difficult.
Here, we present an RCM based on a lognormal distribution (Forney and Rothman,
2012b):
$$g(k, 0) = \frac{1}{k \cdot \sigma \cdot \sqrt{2\pi}} \cdot e^{-(\ln k - \ln \mu)^2 / (2\sigma^2)} \tag{3}$$

where $\ln \mu$ is the mean of $\ln k$, and $\sigma^2$ is the variance of $\ln k$ (Fig.1D$_3$). Parameter $\mu$
determines the mean reactivity of the initial OM bulk mixture and parameter $\sigma$ reflects the
spread of OM components around the mean reactivity.
The lognormal distribution is formed by the multiplicative effects of random variables,
which is commonly observed in nature (e.g., the radioactivity of elements in the crust, the
incubation period of infectious diseases, and ecological species abundance) (Limpert et al.,
2001). In the ocean system, the rates of ocean primary production and biological carbon
export also fit the lognormal distribution (Cael et al., 2018). The degradation of OM in
natural ecosystems is controlled by a network of biologically, physically, and chemically
driven processes (Forney and Rothman, 2014), so the variables raised from such
multiplicative processes are often followed by a lognormal distribution. Forney and
Rothman (2012b) showed that litter bag OM incubation data is indeed best described by a
lognormal distribution of rates.

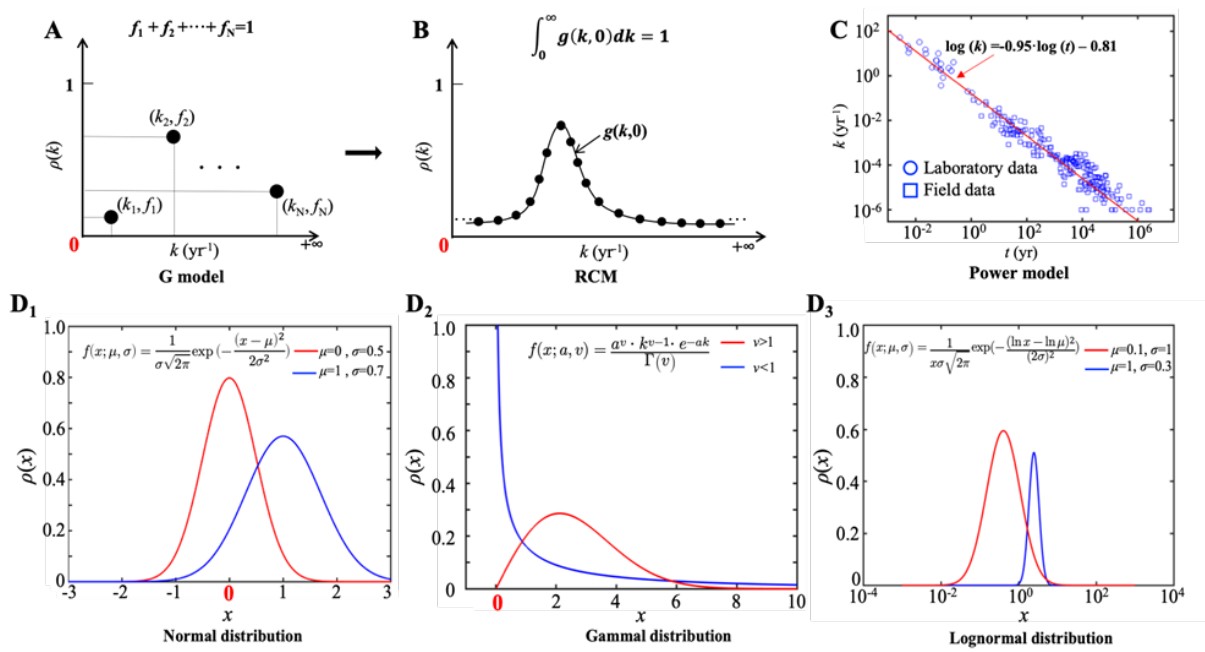

**Figure 1. Schematic diagram of different OM degradation models.** A: G model, B: RCM, C: Power model and D: Common continuum distribution functions. The $x$ coordinate denotes the variation range of values, and the $y$ coordinate denotes the probability density distribution ($\rho$) ($D_1$: the Normal distribution, a typical all-axis distribution, $D_2$: the Gamma distribution, a typical semi-axis ($x>0$) distribution, and $D_3$: the Lognormal distribution, a typical semi-axis ($x>0$) distribution).

In this study, we first compared the $l$-RCM with other OM degradation models and analyzed the advantages of the $l$-RCM in describing the OM reactivity distribution. Then we simulated OM degradation in marine sediment at 123 global sites using the $l$-RCM. Based on inverse modeling results in our dataset, we established the empirical formulas of OM reactivity $vs$ sedimentation rate and further mapped the global OM reactivity distribution. This study provides a new framework for assessing OM reactivity on regional/global scales and predicting OM reactivity in data-poor areas based on easily obtainable environmental parameters (e.g., sedimentation).

## 2 Materials and methods

### 2.1 OM degradation model approach

We constructed an RCM with lognormal distribution ($l$-RCM) to simulate the OM degradation in marine sediments. The $g(k,0)$ we used in Eq. 1 is the lognormal distribution (Eq.3). Because of the tail of $g(k,0)$, the mean rate constant for bulk OM degradation or the apparent degradation rate of the bulk OM ($<k>$) is greater than the median $\mu$, as follows:

$$< k >= \int_0^\infty k \cdot g(k,0)dk = \mu \cdot e^{\sigma^2/2} \tag{4}$$

### 2.2 Inverse model approach

Here, we used 123 published datasets of OM depth profiles across a wide range of
different depositional environments that have been sourced from published literature
(Middelburg, 1989; Arndt et al., 2013; Middelburg et al., 1997) and the IODP database
(Fig.2, Supplementary Table S1) to inversely determine the $\mu$ and $\sigma$ parameters. We also
analyzed a small number ($n=12$) of laboratory experiment data on OM degradation
(Middelburg, 1989), as well as OM degradation data ($n=16$) from terrestrial soils (Katsev
and Crowe, 2015). We followed the inverse modeling approach by Forney et al.(2012a) to
identify the best-fitting parameters $\mu$ and $\sigma$ based on the Newton method.

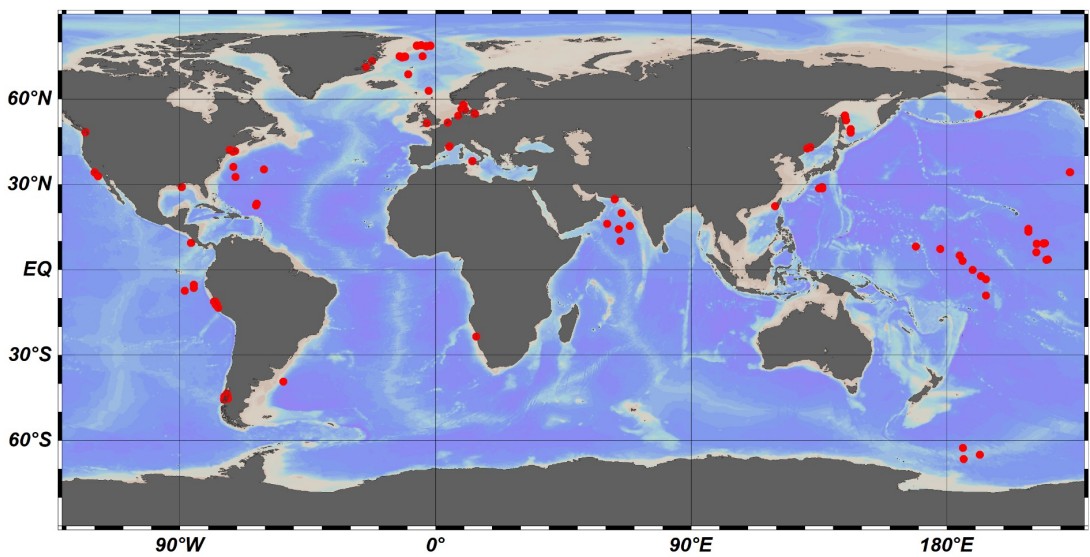

**Figure 2. Global distribution of investigated sites.**

Notably, the burial time was correlated with the porosity. A simple exponential function
was used to describe porosity in sediments:

$$\varphi(x) = \varphi_0 \cdot e^{-\lambda x} \tag{5}$$

where $\varphi_0$ is the values of porosity at the SWI, $\lambda$ is the attenuation coefficient, and $x$ is
depth. Considering the compaction impacts on OM degradation, the burial time
corresponding to each depth in the OM profile can be calculated as:
$$t(x) = \int_0^x \omega^{-1} dx = \frac{x}{\omega_f} + \frac{(\varphi_0 - \varphi_f)}{(1 - \varphi_f) \cdot \lambda \cdot \omega_f} \cdot (e^{-\lambda \cdot x} - 1) \qquad (6)$$

where $\varphi_f$ is the values of porosity at larger depths, calculated from Eq. 5 and the pre-set
simulation depth. If the porosity data were not available, the global set as: shelf regions ($\varphi_0$:
0.45, $\lambda$: $0.5 \times 10^{-3}$), slope regions ($\varphi_0$: 0.74, $\lambda$: $1.7 \times 10^{-4}$), and abyssal regions ($\varphi_0$: 0.7, $\lambda$:
$0.85 \times 10^{-3}$) (LaRowe et al., 2020b).
**2.3 Global upscaling of sedimentation rate**
The inversely determined $\mu$, $\sigma$ couples of all investigated sites were then used in a linear
regression method to derive the empirical relationships between OM parameters $\mu$, $\sigma$, $<k>$
and the local sedimentation rates ($\omega$). A correction factor was applied to account for the
skewness bias inherent in the back conversion from a log-log transformed linear regression
model to arithmetic units. The newly derived empirical relationships between $<k>$ and $\omega$
were then used to calculate global maps of OM reactivity at the SWI on a $1° \times 1°$ grid cell
of the world ocean. At each grid point, $\omega$ was estimated based on the empirical relationship
between $\omega$ ($\omega$ in cm yr$^{-1}$) and the water depth ($z$ in m) (Eq.7, Fig.3), derived from 260
observations on the global continental shelves (Burwicz et al., 2011), complemented here
by an extra 360 sites including abyss regions (data from Arndt et al. (2013), Egger et al.

187    (2018)).

$$\omega(z) = \frac{0.4}{1 + \left(\frac{z}{200}\right)^{3.5}} + \frac{0.004}{1 + \left(\frac{z}{4500}\right)^{17}} \qquad (7)$$

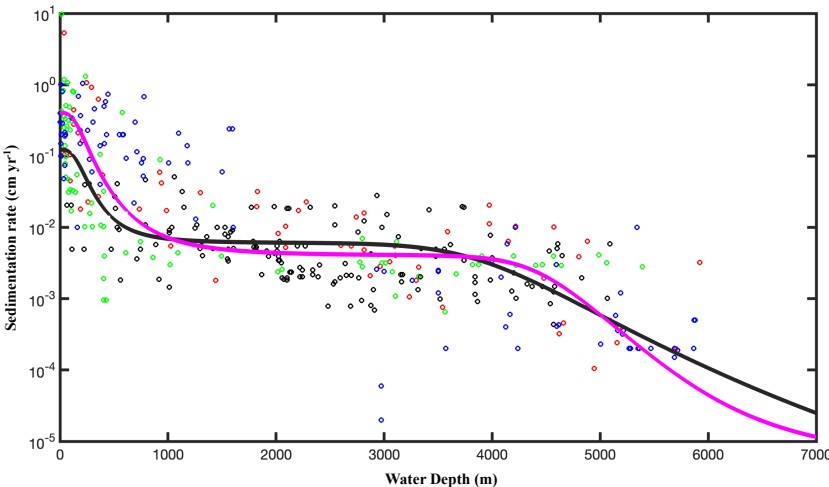


**Figure 3. Relationship between Sedimentation rate (*w*) and water depth (*z* in m).** The data are taken from Arndt et al. (2013) (black circles), Egger et al.(2018) (pink circles), Betts and Holland (1991) (red circles), Colman and Holland (2000) (green circles), and Seiter et al. (2004) (blue circles). The pink line is the fitting result according to Eq. 7 ($R^2$=0.57), and the black line is the fit obtained from the data of Burwicz et al. (2001) ($R^2$=0.43).

196

Considering the geographic differences in depositional environments and to describe the global distribution of sedimentary OM reactivity in more detail, we divided the global ocean into 30 different regions (Table 2, Fig.4) using 5600 single measured data of OM content in global surface sediment (<5 cm sediment depth) and the previously used combined qualitative and quantitative geostatistical methods (Seiter et al., 2004).

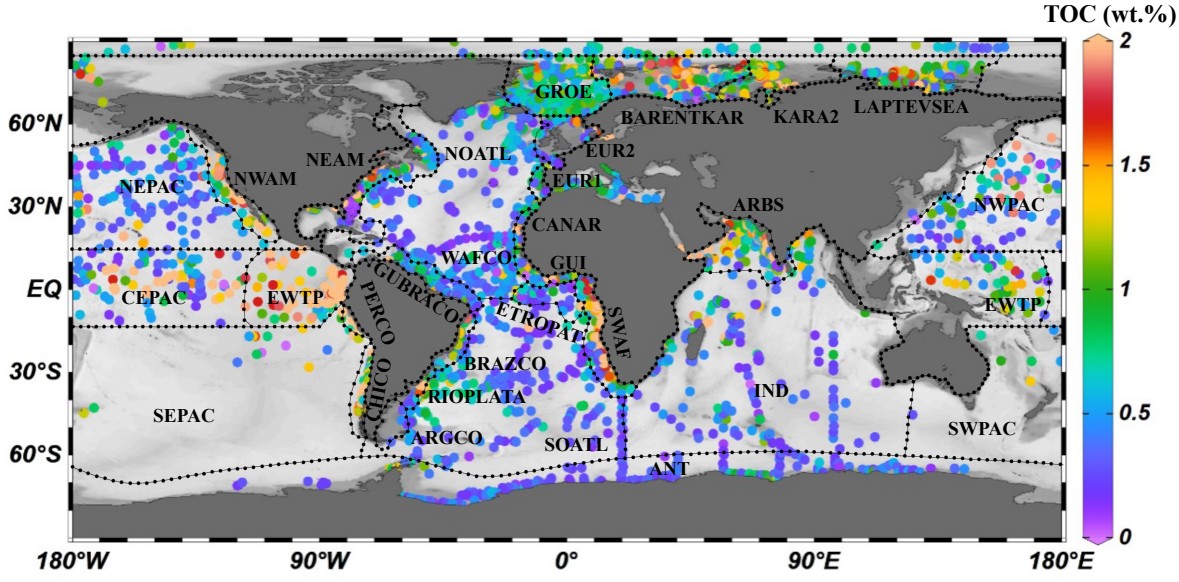

202

**Figure 4. The 30 different regions of the global ocean were divided using 5600 single**

**measured data of OM content (wt.%) of surface sediments**

205

## 3 Results and discussion

### 3.1 OM reactivity distribution described by the $\gamma$-RCM and the $l$-RCM

To compare OM reactivity distribution described by the $l$-RCM and the $\gamma$-RCM, we

determined the best fit to the eight OM datasets reported by Boudreau and Ruddick. (1991).

The results show that both RCMs fit the data equally well, as illustrated by the high

coefficient of determination for each fit ($R^2>0.9$, Table 1 and Fig.5). However, the $l$-RCM

and the $\gamma$-RCM differ in their ability to find a unique solution and in their respective

probability density functions of OM reactivity ($\rho(k)$). For example, Fig.6A and 6B show

the best-fit OM profiles for two contrasting sites: BX-6 on the shelf and DSDP 58 in the

abyssal region. The inversely determined parameters at the two sites are $\mu=2.23\times10^{-3}$ yr$^{-1}$,

$\sigma=2.03$ at BX-6, and $\mu=6.11\times10^{-5}$ yr$^{-1}$, $\sigma=1.66$ at DSDP 58 by the $l$-RCM. At BX-6, the

best-fitting parameters by the $\gamma$-RCM are $v$ =0.278 and $a$=22.5, and at DSDP 58, $v$=1.08
and $a$=20224. According to the parameter sensitivity analysis, the $R^2$ of the fitted results
remains greater than 0.9 when $a$ and $v$ change substantially simultaneously (Fig.6D,
Supplementary Table S2, Fig.S1, S2, and S3). As a result, different combinations of $a$ and
$v$ can fit the data equally well. For example, simultaneously increasing $v$ and $a$ ($v$=0.5 and
$a$=53) at site BX-6 or decreasing $v$ and $a$ ($v$=0.5 and $a$=4024) at site DSDP 58 lead to a
slight change in $R^2$. Adding additional measured data, such as depth profiles of porewater
sulfate and methane concentrations, can help find a unique solution (Freitas et al., 2021).
In contrast, the best-fit parameters $\mu$ and $\sigma$ are unique in the $l$-RCM, and even small changes
in either parameter can lead to abysmal fitting results (Fig.6D). The second difference
between the two models concerns the shape of the probability distribution $\rho(k)$. Statistically,
the features of the Gamma distribution vary with the value of $v$. If $v$<1, $\rho(k)$ tends to positive
infinity when $k$ approaches zero. In contrast, if $v$>1, $\rho(k)$ tends to zero when $k$ approaches
zero. Hence, the characteristics of the Gamma distribution under different $v$ values are
difficult to visually compare the OM reactivity distributions at site BX-6 ($v$<1) and DSDP
($v$>1) (Fig.6C). Compared with $\gamma$-RCM, the $l$-RCM can better distinguish OM reactivity
distribution at different sites.

**Table 1. List of model parameters and coefficients of determination ($R^2$) for the**
**fitting result of $\gamma$-RCM and $l$-RCM.**

| Core | $\gamma$-RCM | | | $l$-RCM | | |
|---|---|---|---|---|---|---|
| | $v$ (-) | $a$ (yr) | $R^2$ | $\mu$ (yr$^{-1}$) | $\sigma$ (-) | $R^2$ |
| Foam | 0.152 | 4.2 | 0.930 | $2.2\times10^{-3}$ | 3.725 | 0.923 |
| SCR-44 | 0.202 | 70.4 | 0.929 | $4.4\times10^{-4}$ | 2.706 | 0.922 |
| BX-6 | 0.278 | 22.5 | 0.929 | $2.24\times10^{-3}$ | 2.031 | 0.936 |
| PC2&TW2 | 0.052 | 0.16 | 0.937 | $5.5\times10^{-5}$ | 6.688 | 0.947 |
| 10141&2 | 0.193 | 10184 | 0.935 | $1.9\times10^{-6}$ | 3.289 | 0.936 |
| 7706-41K | 0.910 | 141.3 | 0.974 | $9.5\times10^{-3}$ | 0.899 | 0.972 |

| | | | | | | |
|---|---|---|---|---|---|---|
| **7706-36** | 0.804 | 231.7 | 0.978 | $4.79\times10^{-4}$ | 1.089 | 0.980 |
| **DSDP58** | 1.080 | 20224 | 0.917 | $6.11\times10^{-5}$ | 1.663 | 0.921 |


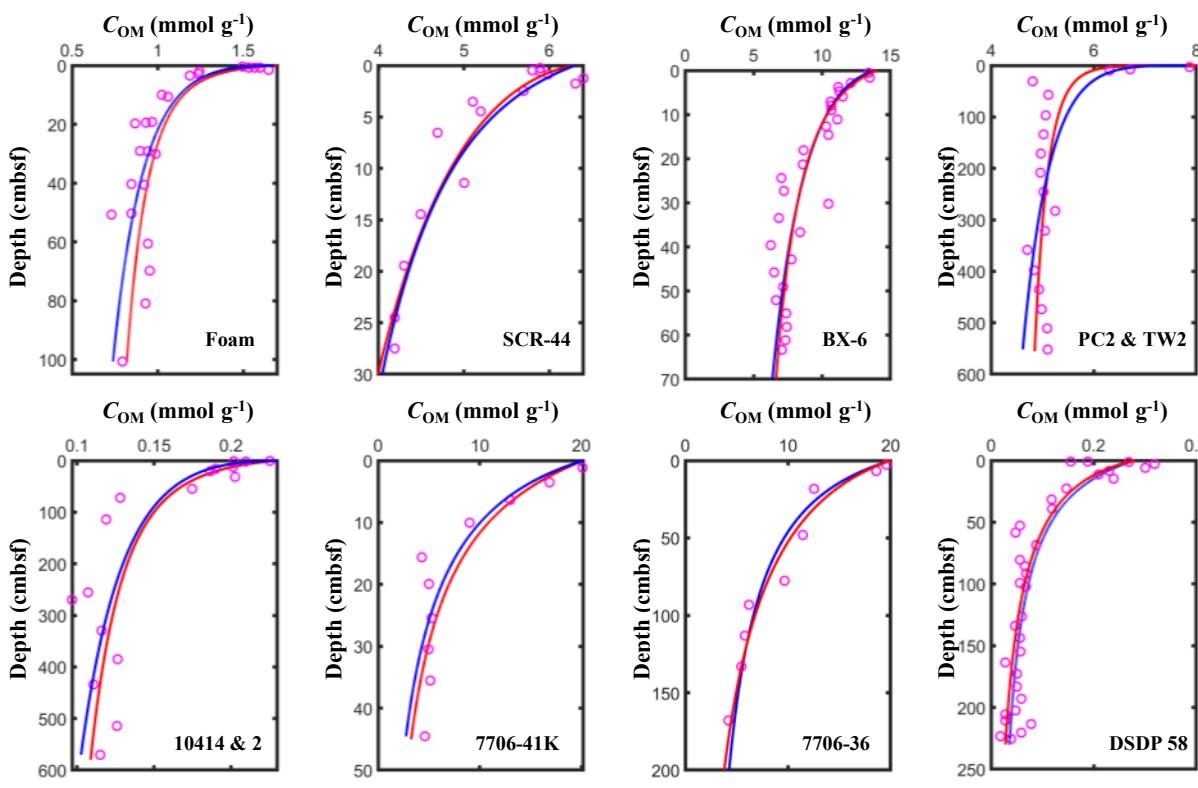

**Figure 5. Fitting results of the *l*-RCM and the *γ*-RCM.** The pink dots are the measured
OM data, the red lines are *l*-RCM fitting results, and the blue lines are *γ*-RCM fitting results.

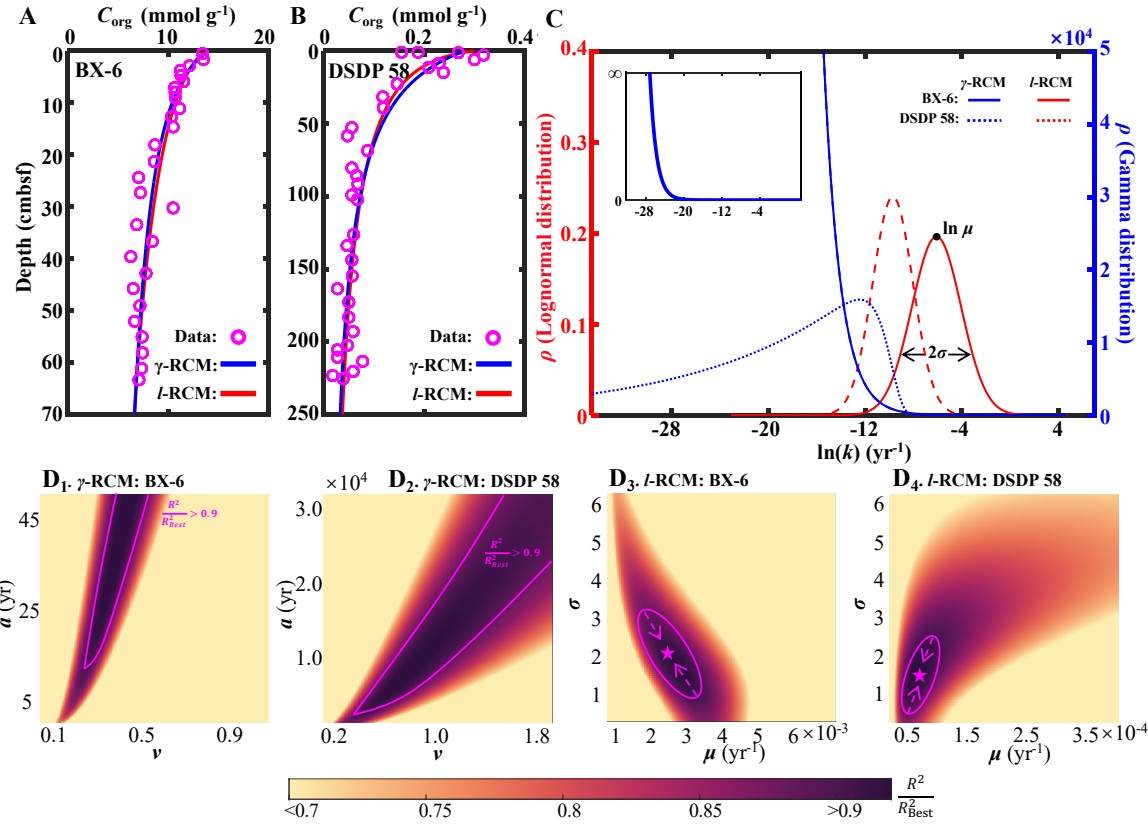

**Figure 6. Comparison of *l*-RCM and *γ*-RCM. A**, **B**: the fitting results of the *l*-RCM and the *γ*-RCM for site BX-6 and DSDP 58. **C**: OM reactivity distribution from *l*-RCM and *γ*-RCM. Top inset, Gamma distribution at site BX-6 at a larger y-axis. **D**: Distribution of $R^2/R^2_{\text{Best}}$ for parameter sensitivity analysis of the *γ*-RCM and the *l*-RCM at sites BX-6 and DSDP 58. The pink lines in the $D_1$ and $D_2$ denote the range that $R^2/R^2_{\text{Best}}>0.9$ in the *γ*-RCM. The $R^2/R^2_{\text{Best}}$ in the *l*-RCM converges as the pink arrows in the $D_3$ and $D_4$, ultimately reaching the best fitting results as the pink pentagrams.

## 3.2 Regional distribution of OM reactivity

In the *l*-RCM, parameter $\mu$ represents the mean reactivity of the OM fractions, which dominates the rate of OM degradation (Supplementary Fig.S2), and parameter $\sigma$ describes the homogeneity of OM fractions, with larger $\sigma$ value indicating more heterogeneous mixture of OM (Forney et al., 2012b). The inverse determination of the *l*-RCM parameters

$\mu$ and $\sigma$ across the wide range of different depositional environments allows quantitative
insights into OM reactivity and provides essential information on the main environmental
controls on OM reactivity. Fig.7 illustrates the inversely determined $\mu$-$\sigma$ for all 123 depth
profiles of marine sediment POC investigated in this study and compares them with
inversely determined parameters from published soil and laboratory incubation data. It
highlights the large inter- and intraregional variability of best-fit $\mu$ ($10^{-6}$–$10^2$ yr$^{-1}$) and $\sigma$
(0.2–6). However, despite the large variability, it also reveals broad global patterns in $\mu$
and $\sigma$. Notably, best-fit $\mu$-$\sigma$ couples form environmental clusters along a $\mu$ gradient, with
the highest $\mu$ being determined for laboratory degradation experiments of fresh
phytoplankton (Garber, 1984; Westrich and Berner, 1984) ($\mu$=$10^0$–$10^2$ yr$^{-1}$), followed by
soil incubation under natural (Katsev and Crowe, 2015), yet still idealized conditions
($\mu$=$10^0$–$10^1$ yr$^{-1}$), while OM degraded in marine sediments generally reveals lower
inversely determined $\mu$<$10^0$ yr$^{-1}$. The higher $\mu$ values determined for soil OM seemingly
contradict the widely accepted notion that soil OM is generally less reactive than marine
OM (Larowe et al., 2020a; Zonneveld et al., 2010). However, this apparent contradiction
can be explained by the idealized conditions of the incubation experiments (e.g., only one
type of material, some of which had nitrogen added), as well as the degradation state of the
investigated OM. Although soil OM is structurally less reactive (Hedges and Keil, 1995;
Zonneveld et al., 2010), the soil incubation experiments were conducted with initially
undegraded material. In contrast, OM deposited in marine sediments consists of a complex
mixture of OM from autochthonous and allochthonous sources that is altered to various
degrees during transit from its source to the sediment (Hewson et al., 2012).
In addition to the difference between incubation data and field observations, Fig.7 also
reveals a three order of magnitude decrease in inversely determined $\mu$ for OM from the
shelf ($10^{-3}$–$10^{-1}$ yr$^{-1}$) to the slope ($10^{-4}$–$10^{-3}$ yr$^{-1}$), and ultimately abyssal regions (<$10^{-4}$ yr$^{-1}$
$^{1}$). In addition, shelf and slope regions also generally reveal a larger $\sigma$ (1–3), while abyssal
regions display a narrower $\sigma$ range (0.5–1). This observed progressive decrease in $\mu$ and $\sigma$
from the shelf to the abyssal ocean confirms previously observed patterns (Arndt et al.,
2013; Freitas et al., 2021; Zonneveld et al., 2010) and reflects the interaction between OM
structure (or its source) and the degree of alteration/pre-processing as OM transits from its
original source to the ultimate sedimentary sink. In the dynamic shelf regions, highly
variable OM loads from different sources, including *in*-situ produced marine OM, laterally
transported, pre-processed terrestrial or marine OM, are often physically protected from
further erosion/deposition cycles due to high suspended sediment loads (Arndt et al., 2013;
Larowe et al., 2020a). As a result, benthic OM is composed of a complex mixture of fresh
and pre-aged compounds of highly variable (hence larger $\sigma$ of the initial distribution), yet
generally higher reactivity. On the upper and mid-continental slopes, intensive lateral
and/or vertical transport processes or the abrupt relocation of sediment result in similar
complex mixtures of OM (hence similar $\sigma$ of the initial distribution) (Larowe et al., 2020a).
However, transport timescales are often longer due to the greater water depths and distance
to land. The deposited OM is generally more degraded and thus less reactive than in shelf
environments. In contrast, benthic OM in abyssal regions is mainly derived from marine
production (Rowe and Staresinic, 1979; Larowe et al., 2020a). During its slow settling
through the water column, highly reactive OM compounds are rapidly degraded, and only
the less reactive compounds persist and settle onto the sediment (Dunne et al., 2007). The
values of $\mu$ and $\sigma$ in the abyssal regions are thus significantly smaller than in the shelf and
slope regions. The decrease of $\mu$ and $\sigma$ from the shelf to abyssal regions reveals a decline
in reactivity during lateral transport of OM, where $\mu$ mainly controls the overall reactivity
and $\sigma$ indicates the coverage of the main component of OM.

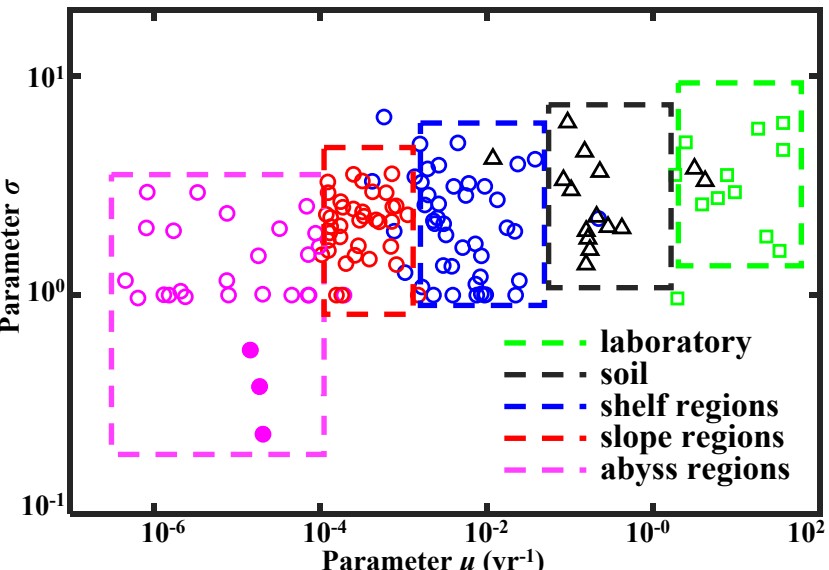


**Figure 7. Regional distribution of OM reactivity.** Distribution of parameters $\sigma$ and $\mu$ in
different regions. Pink solid circles denote fitting results of sites in the NEPAC with
extremely low OM reactivity.

**3.3 Global distribution patterns of OM reactivity**
Parameters $\mu$ and $\sigma$ together control the degradation process of OM, which can be further
described by the apparent degradation rate of the bulk OM ($<k>$). Sedimentation rate ($\omega$)
is a widely observed and comparably easy to measure proxy for local depositional
conditions with sizable global data sets or empirical formulas available (Burwicz et al.,
2011). Fig. 8A, 8B and 8C show the global decreasing trend of $\mu$, $\sigma$ and $<k>$ with $\omega$ for the
general sea regions (shelf (<200m), slope (200–2000m), and abyss (>2000m)). The active
OM fractions (e.g., sugars and proteins) are preferentially exhausted during the lateral
transport of OM from the shelf to the abyssal regions, leading to a decrease in the mean
OM reactivity ($\mu$, Fig. 8A), and thus OM is mainly composed of refractory components ($\sigma$,
Fig. 8B). Due to the multiple sources of OM in the shelf regions, including fresh and older
OM imported laterally by inland rivers, and OM settled from the euphotic layer (LaRowe
et al., 2020a), the values of the values of $\mu$, $\sigma$ and $<k>$ fluctuates significantly. However,
the general trend is superimposed by a large variability and apparent reactivity $<k>$ in
specific environments, notably deviating from this generally observed trend. More
specifically, higher $\mu$ and $\sigma$ values and, thus, higher OM reactivities occur in the Eastern-
Western Coastal Equatorial Pacific (EWEP), Southwestern-Africa continental margin
(SWAF), Northwestern-America continental margin (NWAM), and the Arabian Sea
(ARBS) regions. These results are completely consistent with prior observations and model
results (Arndt et al., 2013) and can be directly linked to the prevailing water-column redox
and depositional conditions. High benthic OM reactivities have previously been reported
for depositional environments that are characterized by a dominance of marine algal OM
(Hammond et al., 1996) and strong lateral transport processes (e.g., SWAF, NWAM)
(Arndt et al., 2013). Consequently, the larger values of all $\mu$ and $\sigma$, and $<k>$ occur in the
inverse modelling results for these depositional environments (Fig. 8). Furthermore, the
reactivity of sedimentary OM is considerably influenced by oxygen content or more
precisely, by oxygen exposure time in the water column and at the seafloor (Aller, 1994;
Hartnett et al., 1998; Hedges and Keil, 1995; Mollenhauer et al., 2003; Zonneveld et al.,
2010). Lower oxygen concentrations, as present in these regions in the form of pronounced
oxygen minimum zones (OMZs), will slow down the degradation of OM both in the water
column and at the sediment surface (Jørgensen et al., 2022). This enables the burial of more
reactive OM into the sediments and thus results in the occurrence of high sedimentary OM
reactivity in these regions despite great water depth (e.g., ARBS, EWTP) (Arndt et al.,
2013; Bogus et al., 2012; Ingole et al., 2010; Luff et al., 2000; Volz et al., 2018). The *l*-
RCM not only captures the broad patterns of OM reactivity across the global seafloor even
better than previous models, but also provides statistically more significant relationships
between OM reactivity ($<k>$) and sedimentation rate ($\omega$) than inversely determined
parameters of $\gamma$-RCM ($R^2<0.46$) and discrete models ($R^2<0.1$) (Arndt et al., 2013).
Considering that no robust quantitative framework exists at this stage to predict OM
reactivity as a function of easily observable environmental parameters, the *l*-RCM provides
an excellent first-order predictor and a step forward in assessing the global distribution
patterns of OM reactivity, despite the poor relationship between $<k>$ and $\omega$ for these special
regions (e.g., EWEP, SWAF, NWAM, and ARBS).

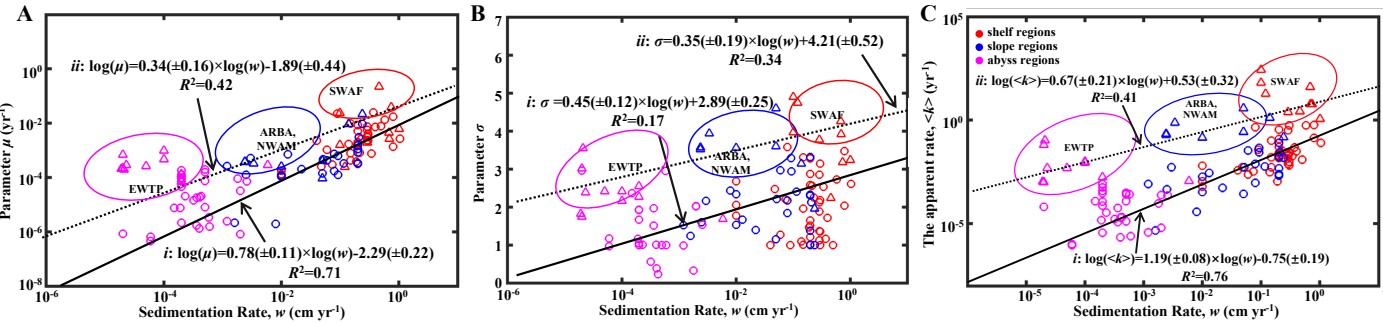

**Figure 8. Global distribution patterns of OM reactivity.** A. Log-log plot of $\omega$ and $\mu$. B.
Log-log plot of $\omega$ and $\sigma$. C. Log-log plot of $\omega$ and $<k>$. The solid black line (*i*) denotes
linear regression for shelf, slope, and abyssal regions. The black dotted line (*ii*) denotes
linear regression for high OM reactivity regions, including the EWTP, ARBS, NWAM,
and SWAF regions.

Based on the empirical relationships in Fig.8 (*i* for the general water depth-related
regions, *ii* for the specific regions (EWTP, ARBS, NWAM, and SWAF)), and the water
depth–$\omega$ relationship (Eq.5), we finally derived, to our knowledge, the world's first map
of the global distribution of parameter $\mu$, $\sigma$, and $<k>$ (Fig.9). Using the relationship between
water depth, $\omega$, and $<k>$ (Fig.3 and Fig.8C), we further estimated the mean apparent OM
reactivity ($<K_{region}>$) in the 30 regions of global ocean (Table 2). Furthermore, the
heterogeneity of the OM reactivity distribution in global marine sediments is well
illustrated in Fig. 9. Specifically, higher $\mu$ (Fig.9A), $\sigma$ (Fig.9B), and OM reactivity (Fig.9C)
is reflected in shelf regions, particularly in northern Atlantic provinces with high latitudes
(e.g., Barents Sea ($<K_{region}> \approx 0.02$ yr$^{-1}$), Laptev Sea ($<K_{region}> \approx 0.03$ yr$^{-1}$), and Kara Sea
($<K_{region}> \approx 0.01$ yr$^{-1}$)), due to shallower water depths and high OM fluxes from inland
(Burwicz et al., 2011; Seiter et al., 2004). Besides that, the global map also highlights the
extremely low OM reactivity, especially in some regions, as indicated by the absence of
sulfate-methane transition (SMT) (e.g., the NE-Pacific, NEPAC) (Eggert et al., 2018) and
central ocean gyre regions (e.g., South Pacific Gyre) (LaRowe et al., 2020b). Deeper water
depth (>5000m), relatively low OM content (~0.2wt.%), and the old OM age (>$10^4$ years)
result in comparably lower $\mu$ and $\sigma$ values (Fig.9A and 9B) and, thus, extremely low benthic
OM reactivity ($<K_{region}> \approx 10^{-4}$ yr$^{-1}$) (Kallmeyer et al., 2012, Müller and Suess, 1979).
Normally, greater water depth enhances oxygen exposure time for OM degradation, and
thereby reduce the reactivity of OM arriving at the seafloor, as reflected in the smaller $\mu$
values (Fig. 9A). In ocean areas characterized by pronounced OMZs, however, due to
strong coastal upwelling or a high export rate of plankton-derived OM, the inhibition of
OM degradation processes in the water colcunm results in the preservation of
heterogeneously mixed OM components (both active and refractory), as reflected in the
larger $\sigma$ values (Fig. 9B), leading to higher than expected OM reactivity in specific regions
despite greater water depths (e.g., ARBS and EWTP ($<K_{region}> \approx 0.01$ yr$^{-1}$) (Fig. 9C). Thus,
the $l$-RCM provides a new framework not only for identifing the differences in OM
reactivity between regions, but also for assessing regional/global OM reactivity patterns
using easily obtainable information (e.g., sedimentation).

OM reactivity exerts an important control on the relative significance of OM degradation

pathways in marine sediments. In oxic environments, OM will be mainly respired
aerobically and through denitrification, whereas deeper within the sediment, it will mainly
be decomposed through anaerobic pathways such as sulfate reduction and methanogenesis
(Regnier et al., 2011). Therefore, further work should be conducted to simulate the
associated biogeochemical processes using the $l$-RCM to better quantify OM degradation
and burial in marine sediments on regional or global scales.

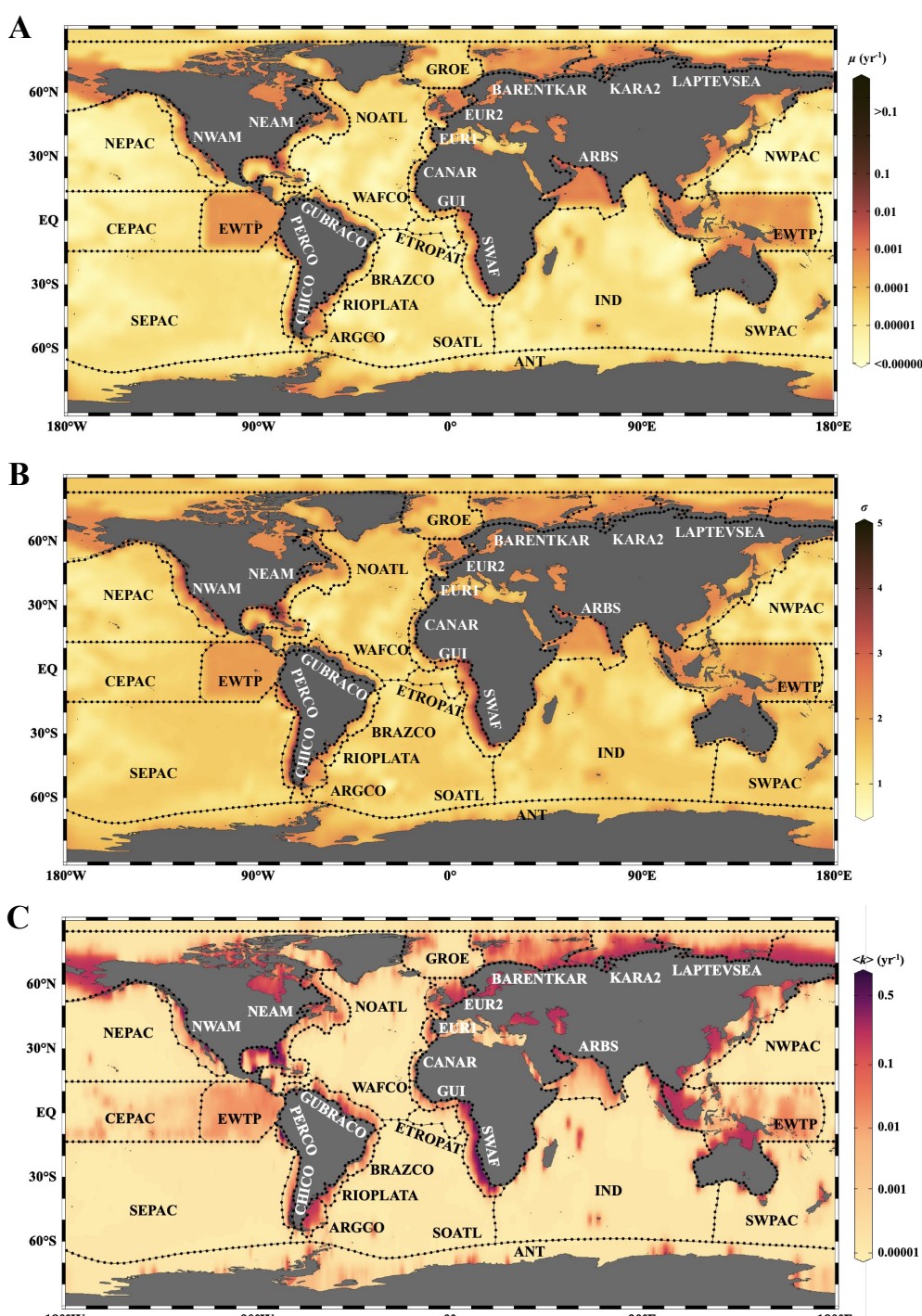

Figure 9. Distribution of $\mu$ (A), $\sigma$ (B), and $<k>$ (C) in the global ocean with 1°×1° resolution.

**Table 2. Abbreviations of regions in this paper (Seiter et al., 2004), and their area, mean water depth, mean OM content in surface sediment (<5 cm), and apparent OM degradation rata (<$K_{region}$>).**

| Abbreviation | Region | water depth[a] (m) | Mean OM (wt.%) | <$K_{region}$> (yr$^{-1}$) |
|---|---|---|---|---|
| SWAF | SW-Africa continental margin | 334 | 2.5 | 0.48542 |
| NWAM | NW-America continental margin | 731 | 1.7 | 0.12695 |
| ARBS | Arabian Sea | 1600 | 1.4 | 0.08182 |
| EWTP | East-West Coastal Equatorial Pacific | 3662 | 1.2 | 0.01587 |
| ANT | South Polar Sea | 1300 | 0.3 | 0.00029 |
| ARGCO | Argentina continental margin | 1859 | 0.3 | 0.00026 |
| BARENTKAR | Barents Sea and Kara Sea | 224 | 1.1 | 0.02081 |
| BRAZCO | Brazil continental margin | 1051 | 0.5 | 0.00034 |
| CANAR | Canaries | 1190 | 0.6 | 0.00031 |
| CEPAC | Central Equatorial Pacific | 5022 | 0.3 | 0.00002 |
| CHICO | Chile continental margin | 1444 | 1.5 | 0.00028 |
| ETROPAT | Eastern tropical Atlantic | 2253 | 0.7 | 0.00026 |
| EUR1 | N-European continental margin | 1290 | 0.8 | 0.00029 |
| EUR2 | S-European continental margin | 974 | 0.3 | 0.00037 |
| GROE | Northern Nordic Sea | 1563 | 0.7 | 0.00027 |
| GUBRACO | SE-America continental margin | 1844 | 0.4 | 0.00026 |
| GUI | Gulf of Guinea | 1586 | 1.1 | 0.00027 |
| INA | Indian Ocean deep sea | 4042 | 0.4 | 0.00021 |
| KARA2 | Kara Sea | 281 | 1.2 | 0.01111 |
| LAPTEVSEA | Laptev Sea | 190 | 0.9 | 0.02964 |
| NEAM | NE-America continental margin | 1045 | 0.9 | 0.00034 |
| NEPAC | NE-Pacific | 4463 | 0.4 | 0.00012 |
| NOATL | Northern Atlantic | 2161 | 0.4 | 0.00026 |
| NWPAC | NW-Pacific | 4898 | 0.6 | 0.00004 |
| PERCO | Peru continental margin | 1020 | 4.8 | 0.00035 |
| RIOPLATA | Rio de la Plata mouth | 1784 | 0.8 | 0.00026 |
| SEPAC | SE-Pacific | 3952 | 0.5 | 0.00022 |
| SOATL | Southern Atlantic | 3592 | 0.4 | 0.00024 |
| SWPAC | SW-Pacific | 3153 | 0.8 | 0.00025 |
| WAFCO | W-Africa continental margin | 1982 | 0.6 | 0.00026 |

[a]water depth and mean OM content are based on the average depth and OM content of the sites in each region of Fig.4.

# 4 Conclusions

Compared with previous OM degradation models, the *l*-RCM presented here not only

well fits OM depth-content profiles, but also better represents the distribution of OM
reactivity by the parameters $\mu$ and $\sigma$. We use the *l*-RCM to inversely determine $\mu$ and $\sigma$ at
123 sites across the global ocean, including shelf, slope, and abyssal regions. Our results
show that the apparent OM reactivity ($<\!k\!>=\mu\cdot\exp(\sigma^2/2)$) decreases with decreasing
sedimentation rate ($\omega$), and that OM reactivity is more than three orders of magnitude
higher in shelf than in abyssal regions. Due to the complex depositional environments (e.g.,
oxygen minimum zones), OM reactivity is higher than predicted in some specific regions
(e.g., the NWAM, SWAF, ARBS, and EWTP), which was also captured by the *l*-RCM in
these regions. Based on two empirical relationships between the OM reactivity ($<\!k\!>$) and
sedimentation rate ($\omega$), we obtained the global OM reactivity distribution patterns and
finally mapped the global OM reactivity distribution. The reactivity of OM serving as fuel
for microbial activity in marine sediments firmly controls the degradation pathways and
metabolism rates. Thus, the *l*-RCM has direct implications on the constraints for OM
degradation and burial in marine sediments on regional or global scales.

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

**Acknowledges**

This study was supported by National Key Basic Research and Development Program of China (2022YFC2805400; 2016YFA0601100), the Natural Science Foundation of China (41976057; 42276059). Sinan Xu gratefully acknowledges the financial support by the China Scholarship Council (contract N. 201906260048) for a research stay at AWI, Germany. Bo Liu and Sabine Kasten acknowledge the BMBF MARE: N project "Anthropogenic impacts on particulate organic carbon cycling in the North Sea (APOC)" (03F0874A).

**Author contributions**

S.X. and B.L. designed the study and performed the research with S.A., S.K., and Z.W.; All authors discussed the results and commented on the manuscript.

**Competing interests**

The authors declare that they have no competing interests.