# Peer review of "Assessing global-scale organic matter reactivity patterns in marine"

_Biogeosciences, 2022_

## Author Comment (AC1)

Dear editor, dear referee, dear EGU sphere,

First, we would like to thank the referee for the time invested reviewing the paper, and the constructive comments and suggestions that were made. Here, we answer in detail some of the most important comments. We took notes of the other suggestions regarding language, or small additions.

**Question 1:** The mathematics is rather complex to understand, including terms as "gamma"-function and "lognormal"-function. According to Wikipedia: "In probability theory, a log-normal (or lognormal) distribution is a continuous probability distribution of a random variable whose logarithm is normally distributed. Thus, if the random variable X is log-normally distributed, then Y = ln(X) has a normal distribution". It should be thought through what physical meaning this has, i.e., what in the nature of a mix of organic matter gives rise to this function?

**Response:** Thanks for your valuable comments and suggestions. Based on the decay equation (Eq.1), reactive continuum models (RCMs) assume that OM compounds are continuously distributed over a wide range of reactivities and characterize the distribution of organic matter reactivity by a continuum distribution function.

$$\frac{dG}{dt} = -kG \quad \rightarrow \quad G(t) = G(0) \cdot e^{-kt} \tag{1}$$

Therefore, different continuum distribution functions can be selected to construct different RCMs. Statistically, the common continuum distribution functions include normal distribution functions, uniform distribution functions, exponential distribution functions, lognormal distribution functions, Kersey distribution functions, Gamma distribution functions, Rayleigh distribution functions, and Weber distribution functions. Considering the degradation process of organic matter, the $k$ value in Eq. 1 must be greater than 0 ($k > 0$), the distribution function used to construct the RCMs is preferably located on a positive semi-axial (Fig. 1, normal distribution (all-axial distribution, $x \in (-\infty, +\infty)$), and lognormal distribution (semi-axial distribution, $x \in (0, +\infty)$)). Hence, some of the continuum distributions mentioned above are not appropriate for constructing RCM (e.g., normal distribution functions, uniform distribution functions, Kersey distribution functions, exponential distribution functions).

[Figure]

**Figure 1. Normal distribution and lognormal distribution**

Theoretically, lognormally distributed variables arise naturally from multiplicative processes (Limpert 2001). In other words, the process is generated by multiple variables in concert. The degradation of organic matter in marine sediments is controlled by several factors (e.g., the presence of electron acceptors, the presence of microbe, the conditions of hydrolytic enzymes), which coincides with the generation of lognormal distribution.

In fact, the lognormal distribution is widely observed in nature (e.g., the radioactivity of elements in the crust, the incubation period of infectious diseases, ecological species abundance). Some oceanographic studies show that the rates of ocean primary production and biological carbon export in the global ocean also follow the lognormal distribution. Moreover, inverse modeling of OM degradation data from 27 different types of litter, including leaves, wood, grass, and wheat, distributed across North America, ranging from the Alaskan tundra to the Panamanian rainforests, showed that OM reactivity followed a lognormal distribution. Based on the above, the lognormal distribution is suitable for characterizing the distribution of organic matter reactivity in marine sediments.

**Question 2:** It is understandable that microbial activity and abundance is ultimately controlled by the rate of decay of larger molecules via hydrolysis, then giving rise to a cascade of fermentation processes. While a single compound reacts with an exponential decay function (Berner, 1964), it has been the common tenet that a complex mixture of compounds results in an approximate power law decay function (see Tarutis, 1993). We

can therefore ask the heretical question why it matters, whether a gamma function or a lognormal function is used, if the physical meaning is anyway the one of power-law functions.

**Response:** Thank you for your suggestion. Here, we will elaborate on the differences between the G model and RCMs, and the advantages of RCM based on lognormal distribution (*l*-RCM) over the commonly used RCM based on Gamma distribution (*γ*-RCM).

In order to express the meaning of organic matter degradation model more clearly, we supplemented the schematic diagram, as shown in Figs.2- 4. Discrete models divide the bulk OM pool into a number of discrete fractions, and each fraction has its own constant reactivity. According to the decay equation, the OM degradation model can be expressed by Equation 1, and the schematic diagram is shown in Fig.1.

[Figure]

$$G(t) = f_1 \cdot G(0) \cdot \exp(-k_1 t) + f_2 \cdot G(0) \cdot \exp(-k_2 t)$$
$$(f_1 + f_2 = 1)$$

$$G(t) = f_1 \cdot G(0) \cdot \exp(-k_1 t) + f_2 \cdot G(0) \cdot \exp(-k_2 t) + f_3 \cdot G(0) \cdot \exp(-k_3 t)$$
$$(f_1 + f_2 + f_3 = 1)$$

**Figure 2. Schematic diagram of multi-G modes (2G-model and 3-G model)**

Based on the Fig.1, we can map the reactivity of organic matter (*k*, *x*-axis) and its fraction (*f*, *y*-axis), which is shown in the Fig. 3.

[Figure]

**Figure 3. Distribution of OM reactivity (*x*-axis) and their fraction (*y*-axis) in G-models**

Continuum model is an alternative to discrete model. Based on the assumption that OM compounds are continuously distributed over a wide range of reactivities, the reactive continuum models (RCMs) can be described by a continuous distribution function (Fig. 3).

[Figure]

**Figure 4. Distribution of OM reactivity (*x*-axis) and their fraction (*y*-axis) in RCMs**

The reactivity of organic matter showed an exponential decrease with time. However, The G model assumes that organic matter consists of a finite number of reactive components, and therefore cannot characterize the decline in organic matter reactivity ($k(t)$) over time. As two horizontal lines at the earlier and later stages in multi-G model (Fig.5), the value of which is related to the choose of $k$ in multi-G model. In the power model, an empirical relationship derived from a large number of datasets shows $k(t) \sim t^{-1}$ ranging from over 8 orders of magnitude of apparent reactivity (Middelburg, 1989). By setting $v=0.125$ and $a=0.0003$ in the $\gamma$-RCM (Boudreau et al., 2008) or $\mu=0.3$ yr$^{-1}$ and $\sigma=6$ in the $l$-RCM, $k(t)$ shows a similar result (Fig.5).

[Figure]

**Figure 5. The relationship of organic matter reactivity ($k(t)$) and time in different models. The data of laboratory experiments and field are summarized by Middelburg, (1989). The parameters 2-G model ($k_1$=2×10⁻¹ yr⁻¹, $f_1$=0.68; $k_2$=3×10⁻⁴ yr⁻¹, $f_2$=0.32) are from (Luff et al., 2004). $k(t)$=0.16$t$-0.95 in power model (Middelburg, 1989) and $k(t)$=0.125/(0.0003+$t$) in gamma RCM (Boudreau et al., 2008).**

The key to RCMs is to select an appropriate distribution function. At the current stage, the $\gamma$-RCM is usually used, where $g(k,0)$ in the Fig.4 is the Gamma distribution function (Eq.2).

$$g(k, 0) = \frac{a^v \cdot k^{v-1} \cdot e^{-ak}}{\Gamma(v)} \tag{2}$$

However, we found that the Gamma distribution function itself has some disadvantages in describing organic matter distribution.

For example, when $v$ is less than 1, the Gamma function tends to +∞ at zero ($x \rightarrow 0$), and when v is greater than 1, the gamma function tends to 0 at zero (Fig. 6). Given that the degradation process of organic matter simulated by RCM using Gamma function in two sites, the best simulation results at one site $v$ values greater than 1 and the other one less than 1. Therefore, it is not feasible to use it to compare organic matter reactivity within these two sites. We have also discussed other disadvantages in Gamma function in detail in the **Section 3.1.**

[Figure]

**Figure 6. Gamma distribution function under different values of parameter $v$**

In $l$-RCM, two parameters ($\mu$, $\sigma$) are used to describe the process of OM degradation. The position of the peak point $\ln(\mu)$ is the most important factor for controlling its distribution range, and $\sigma^2$ is the variance of $\ln k$. Fig.7 shows that the lognormal distribution does not exhibit different distribution characteristics as the gamma distribution when the parameters are taken with different values. Therefore, we can better compare the differences in organic matter reactivity at different sites by using parameters $\mu$ and $\sigma$ in the same axis.

[Figure]

**Figure 7. Schematic diagram of lognormal distribution**

Based on the above comparative analysis, we constructed the RCM using a lognormal distribution function.

Next, we will reorganize the above relevant description and add it to the main text of the revised manuscript. In addition, we will also fully consider other minor issues proposed by the Reviewer in the revised version.

**Preference**

Boudreau, B. P., Arnosti, C., Jørgensen, B. B., and Canfield, D. E., 2008, Comment on" Physical model for the decay and preservation of marine organic carbon": Science, v. 319, no. 5870, p. 1616-1616.

Limpert, E., Stahel, W. A., and Abbt, M., 2001, Log-normal distributions across the sciences: keys and clues: on the charms of statistics, and how mechanical models resembling gambling machines offer a link to a handy way to characterize log-normal distributions, which can provide deeper insight into variability and probability—normal or log-normal: that is the question: BioScience, v. 51, no. 5, p. 341-352.

Luff, R., Wallmann, K., and Aloisi, G., 2004, Numerical modeling of carbonate crust formation at cold vent sites: significance for fluid and methane budgets and chemosynthetic biological communities: Earth and Planetary Science Letters, v. 221, no. 1-4, p. 337-353.

Middelburg, J. J., 1989, A simple rate model for organic matter decomposition in marine sediments: Geochimica et Cosmochimica acta, v. 53, no. 7, p. 1577-1581.

---

## Author Comment (AC2)

Dear editor, dear referee, dear EGU sphere,

First, we would like to thank the referee for the time invested reviewing the paper, and the constructive comments and suggestions that were made. We take this opportunity to answer the most important questioning and suggestions in detail. We took notes of other minor comments to correct and improve the manuscript in its revised form.

**Question 1:** When all is said and done, the main take home point of this work seems to be Fig. 3 and its subsequent discussion. Perhaps I'm being unduly harsh, but I'm not really sure I see a lot here that is really that new, as is indicated by the discussion toward the latter part of section 3.2. In some senses though, this consistency between the model results here and wide range of diverse observations regarding organic matter reactivity and composition is re-assuring, and in some ways this work does act to help "unify" these observations. On the other hand, in other places (lines 339 and 359), the authors note that "the *l*-RCM can be further used to calculate the budget of OM degradation at regional or global scales and assess the significance of the sedimentary carbon cycle on the hydrosphere and atmosphere." To me at least, adding such a calculation to this manuscript would be as (if not more) important and interesting as is Fig. 3. It could then be compared to other regional and global estimates of such quantities cited on lines 328-330, or reported more recently in Jørgensen et al. (2021, Earth-Sci. Rev. 228:103987). These estimates might also be a way of somewhat independently verifying how "good" this lognormal approach is, as compared to other models of sediment OM reactivity.

**Response:** Thank you for your valuable comments and suggestions. It is really true as the Reviewer suggested that we should add the further application of the *l*-RCM to the revised manuscript. We have read the references you provided. Next, we will simulate the degradation process of organic matter using the *l*-RCM on basis of the distribution characteristics of organic matter reactivity in different regions of the global ocean, estimate the amount of organic matter degradation in global sediments and then compare to regional and global estimates of such quantities using other models, thereby further reflecting the advantages of the *l*-RCM. In the revised version, we will fully

consider the Reviewer's comments and add this part to the section of Discussion.

**Question 2:** The overall manuscript is chopped up in such a way that makes it very hard for the reader (or at least me) to follow. Specifically, things discussed in the Supplementary Material section are not well-referenced in the text, and I was very confused when I first started reading the main text, until I realized I had better go through the Supplementary Material section first.

**Response:** Thank you for pointing out this question. In the revised the manuscript, we will reorganize the structure of the manuscript and put the relevant contents in the Supplementary Material section into the text to improve the readability of the manuscript.

**Question 3:** The math in the supplementary section is very dense and confusing in places (also see point 5 below).

**Response:** We will check and correct this section according to the Reviewer's comment in the revised version.

**Question 4:** Maybe I'm missing something, but there seem to be two definitions of (eqn. 4 and eqn. 7, which is the same as eqn. S3) and in plots like Figs. 2 and 3 it's not clear which is being used. This confusion needs to be cleared up in the revisions.

**Response:** Thank you for this suggestion. We will carefully check and clear up this confusion accordingly.

**Question 5:** The referencing in the early part of the manuscript needs to be cleaned up. You don't write "… Washington and Jefferson (Washington and Jefferson, 1776) said …" but rather "… Washington and Jefferson (1776) said …". Also references with 2 authors do not use et al. (e.g., see lines 138 and 179), and again remove the author names from inside the parenthetical statement.

**Response:** Thank you for pointing put this mistake. In the revised version, we will

check and revise these references accordingly.

**Question 6:** The quantity ρ(ln(k)) or ρ(k) is plotted in several places (Figs.1C, S4, S5, S12, S13). This parameter is not clearly defined in the text (maybe I missed it), and it is also not clear how it relates to other parameters being looked at here (this comment may actually be a specific example of the general concern noted in point 3 above).

**Response:** Thank you for your suggestion. We will revise the definition of $\rho(\ln k)$ in the figure caption. The lognormal distribution is symmetric in the logarithmic coordinate system, as shown in Figure 1.

[Figure]

**Figure 1. Schematic diagram of lognormal distribution**

In this manuscript, the *x*-axis represents the distribution of organic matter reactivity (*k*, yr$^{-1}$). Considering that the coordinate system is a logarithmic coordinate system, the value of the horizontal coordinate is (ln *k*). The *y*-axis represents the fraction corresponding to the organic matter reactivity, which is usually referred as probability density function (PDF) in statistics. In this study, $\rho(\ln k)$ is used to represent the organic matter reactivity. We will give the clear definition of this parameter and address this issue in the revision.

In addition, we will also make further revisions to other minor issues that the Reviewer proposed to improve the readability of this manuscript.

***Thank you very much for your valuable comments and suggestions again.***

---

## Author Response (AR1)

<h1 style="text-align:center">Responses to Reviewers' comments</h1>

Firstly, we wish to express our gratitude to the two reviewers and the responsible editor for their suggestions and comments that helped us to improve our manuscript. We have addressed all comments point-by-point as outlined in detail below.

**Reviewer 1**

**Question 1:** The mathematics is rather complex to understand, including terms as "gamma"-function and "lognormal"-function. According to Wikipedia: "In probability theory, a log-normal (or lognormal) distribution is a continuous probability distribution of a random variable whose logarithm is normally distributed. Thus, if the random variable X is log-normally distributed, then Y = ln(X) has a normal distribution". It should be thought through what physical meaning this has, i.e., what in the nature of a mix of organic matter gives rise to this function?

**Responses:** Thank you for your valuable comments. Natural organic matter consists of a complex and dynamic mix of OM that is derived from different sources, is protected by different degrees/types of mineral associations, and has been exposed to different degrees of degradation. As a result, components of this OM mixture are continuously distributed over the reactivity spectrum. The mathematical form of this initial distribution, $g(k,0)$, cannot be inferred by observations. In the past, different mathematical forms have been used, and statistically, the following common continuum distribution functions could be used: Gamma distribution, Lognormal distribution, and Normal distribution. Here we choose the lognormal distribution because:

(1) Considering that the $k$ value in Eq. 1 must be greater than zero ($k > 0$, Fig.1), the distribution used to construct continuum models of organic matter degradation is preferably located on a positive semi-axial. Hence, some of the all-axial distributions ($x \in (-\infty, +\infty)$) mentioned above are not appropriate for constructing a continuum model (e.g., normal distribution functions), while the lognormal distribution fulfills this criterion (**Lines 92-95**).

(2) The lognormal distribution is formed by the multiplicative effects of random

variables, which is commonly observed in nature (e.g., the radioactivity of elements in the crust, the incubation period of infectious diseases, and ecological species abundance) (Limpert et al., 2001). In the ocean system, the rates of ocean primary production and biological carbon export also fit the lognormal distribution (Cael et al., 2018). The degradation of bulk OM in natural ecosystems can be considered a continuum of individual compound degradation rates controlled by a network of biologically, physically, and chemically driven processes, so the variables raised from such multiplicative processes are often followed by a lognormal distribution. Forney and Rothman (2012) showed that litter bag OM incubation data is indeed best described by a lognormal distribution of rates. By assuming that OM is distributed among a continuous network of states that transform with stochastic, heterogeneous kinetics, Forney and Rothman (2014) found that the degradation rates are approximately lognormal through a complex degradation network **(Lines 119-128).**

In summary, the *l*-RCM is thus mathematically, as well as conceptually suitable for simulating the OM degradation in sediments.

[Figure]

**Figure 1. Schematic diagram of different OM degradation models.** A: G model, B: RCM, C: Power model and D: Common continuum distribution functions. The *x* coordinate denotes the the variation range of values, and the *y* coordinate denotes the probability density distribution ($\rho$) ($D_1$: the Normal distribution, a typical all-axis distribution, $D_2$: the Gamma distribution, a typical semi-axis ($x>0$) distribution, and $D_3$: the Lognormal distribution, a typical semi-axis ($x>0$) distribution).

References:

Forney, D. and Rothman, D.: Inverse method for estimating respiration rates from decay time series, Biogeosciences, 9, 3601-3612, doi:10.5194/bg-9-3601-2012, 2012.

Forney, D. C., and Rothman, D. H.: Carbon transit through degradation networks, Ecological Monographs, 84(1), 109-129, doi: 10.1890/12-1846.1, 2014.

**Question 2:** It is understandable that microbial activity and abundance is ultimately controlled by the rate of decay of larger molecules via hydrolysis, then giving rise to a cascade of fermentation processes. While a single compound reacts with an exponential decay function (Berner, 1964), it has been the common tenet that a complex mixture of compounds results in an approximate power law decay function (see Tarutis, 1993). We can therefore ask the heretical question why it matters, whether a gamma function or a lognormal function is used, if the physical meaning is anyway the one of a power-law function.

**Responses:** Thank you very much for the comment. The Power model is derived from an empirical equation describing the OM degradation rate as a function of time in global marine sediments and it reflects the decrease in OM reactivity with time on a long-time scale. Therefore, the distribution of OM reactivity at a given site cannot be simulated by the Power model. In contrast, it can be described by the RCMs using various distribution functions.

The most used model at this stage is the $\gamma$-RCM. Here, we used lognormal distribution to construct RCM ($l$-RCM). As stated in the text (**Lines 207- 232**), both the best-fit solutions using the gamma and the lognormal function fit the data equally well, and from this point of view, it does indeed not really matter if a gamma or lognormal function is used. However, the $l$-RCM and the $\gamma$-RCM differ in (1) their ability to find a unique solution and (2) in their respective probability density functions of OM reactivity ($\rho(k)$).

**Minor comments:**

**Question 3:** Line 31: The results show that …

**Responses:** Thank you for pointing out this mistake. We have corrected it accordingly. Please see this change in **Line 23**.

**Question 4:** Lines 45-49: I suggest to modify the sentence as follows, thereby specify the references with respect to the listed phenomena:

"In particular, the reactivity of benthic OM imposes a substantial control on the magnitude of benthic carbon export and burial (… sequestration happens in the photic zone!) over geological timescales due to the recycling of inorganic carbon by dissimilatory microbial activity in the deep biosphere (Boudreau, 1992), the dissolution and precipitation of carbonates (Meister et al., 2022), and the production of methane (Dickens et al., 2004).

**Responses:** Thank you for the suggestion. We have modified this sentence accordingly. Please see this change in **Lines 43-48**.

**Question 5:** Line 60: Here it would be good to refer to the power law function (see comments above).

**Responses:** Corrected as suggested.

**Question 6:** Line 92: "Boudreau and Ruddick" is duplicated.

**Responses:** We have checked and revised it accordingly (**Line 96**).

**Question 7:** Line 98: "Middelburg" is duplicated.

**Responses:** Corrected (**Line 101**).

**Question 8:** Line 127: Consider re-organizing the methods description to start with explaining what was simulated.

**Responses:** We have added a sentence to explain our model (**Lines 148-149**).

**Question 9:** Line 142: Title 2.3 should be rephrased: not the sedimentation rate is upscaled, the model is.

**Responses:** We have rephrased the Title 2.3 as "Global upscaling of sedimentation rate" (**Line 176**).

**Question 10:** Line 154: Eq. 5 defines the sedimentation rate as a function of water depth z. However, sedimentation rate has also been observed to vary with depth due to compaction. This has an effect also on the organic matter decay with depth.

**Responses:** Thank you very much for the comment. Indeed, the compaction can impact OM degradation. The degree of compaction is mainly reflected by changes in sediment porosity, which in turn affects the burial time of OM. In the revised manuscript, we have added a description of the impact of porosity changes over burial time during OM decay accordingly (**Lines 165-175**).

**Question 11:** Line 170: Why is a multi-G method used if the log-normal method would be better?

**Responses:** Thank you for raising this issue. Due to the complexity and unclear expression of the calculation method used in the previous manuscript, we have reorganized the structure of the manuscript and used a relatively concise method to calculate the $<K_{regions}>$ in the different regions. In the revised method, $\omega$ was estimated based on the empirical relationship between $\omega$ ($\omega$ in cm yr$^{-1}$) and the water depth ($z$ in m) (Eq.7, Fig.3), and the values of $<K_{region}>$ in the 30 regions (Table 2) were calculated according to the relationship between water depth, $\omega$, and $<k>$ (Fig.3 and Fig.8C). Please see the detailed modifications in Line 171-196 and the Section of 3.3 "Global distribution patterns of OM reactivity".

[Figure]

Figure 3. Relationship between Sedimentation rate (*w*) and water depth (*z* in m).

[Figure]

Figure 8. Global distribution patterns of OM reactivity.

**Question 12:** Line 185: on the shelf,

**Responses:** Corrected as suggested.

**Question 13:** Line 194: Also, Meister et al. (2013) evaluated the effects of a and nü in the reactive continuum model on the sulphate and methane concentration profile. Presumably, the log-normal model has similar effects?

**Responses:** Yes, the *l*-RCM would have a similar effect. The effect of OM model's parameters on methane-sulfate dynamics is primarily controlled by the amount of reactive material that reaches the methanogenic zone (Regnier et al., 2011). Organic matter with high reactivity can prompt its consumption in the upper sediment and thus limits the supply of reactive OM to the methanogenic zone. As a result, methanogenesis is substrate-limited and thereby the deep SMTZ occur. Similarly, the low OM reactivity of the sediment itself can also limit the methane-production process, leading to the occurrence of deep SMTZ.

In the $\gamma$-RCM, OM reactivity is controlled by parameters $a$ and $v$. The rate of methane production and SMTZ depth are very sensitive to parameter $a$ that controls OM reactivity (lower $a$, higher initial reactivity), as well as the decrease of OM reactivity with depth/age (lower $a$, rapid decrease of OM reactivity). The highest methane production rate and shallowest SMTZ depth are thus simulated for intermediate $a$ value that optimizes the amount and reactivity of OM that reaches the methanogenic zone. In the $l$-RCM, the evolution of reactivity with time/age is mainly controlled by parameter $\mu$ (the larger the $\mu$, the slower the decrease), and one would expect a similar pattern as in the $\gamma$-RCM.

Reference:

Regnier, P., Dale, A. W., Arndt, S., LaRowe, D. E., Mogollón, J., and Van Cappellen, P.: Quantitative analysis of anaerobic oxidation of methane (AOM) in marine sediments: A modeling perspective, Earth-Science Reviews, 106(1-2), 105-130, doi:10.1016/j.earscirev.2011.01.002, 2011.

**Question 14:** Line 198: WHAT is divergent?

**Responses:** In Gamma distribution function, if $v<1$, $\rho(k)$ tends to positive infinity when $k$ approaches zero. Conversely, if $v>1$, $\rho(k)$ tends to zero when $k$ approaches zero (Fig. 1D$_3$, **Lines 228-232**).

**Question 15:** Line 242: In the reactive continuum model, the parameter a has actual meaning, as the "initial age". Which parameter would represent this property in the lognormal model?

**Responses:** We have added a schematic diagram to explain the parameters in the $l$-RCM, where ln $\mu$ is the mean of ln $k$, and $\sigma^2$ is the variance of ln $k$. Parameter $\mu$ determines the dominant reactivity of the initial OM bulk mixture, and parameter $\sigma$ determines the spread of OM components around this mean reactivity. Small $\sigma$ indicates a very homogenous mixture, and large $\sigma$ indicates a heterogenous mixing (Fig. 1D$_3$).

**Question 16:** Line 262: Perhaps also refer to the South Pacific Gyre, as the region that is most depleted in organic carbon (see also Kallmeyer et al., 2012).

**Responses:** We have added this region and its corresponding literature in the revised manuscript (**Line 374**).

**Question 17:** Line 287: But often the OMZ is in shallower depth, on the shelf or shelf slope, and also affects anoxic shelf basins.

**Responses:** Yes, we agree that the OMZ often occurs in shallower depth. However, the link between OMZs in coastal ocean and OM reactivity in the underlying sediment is more complex than in open ocean because of the dynamic and heterogeneous nature of the coastal environment. Here, complex mixtures of OM sources in a dynamic environment and other controls, such as OM composition, mineral protection, and temperature, can mask the influence of pelagic sub/anoxia on benthic reactivity. The OMZs mainly occur in the Eastern Equatorial Pacific, the Arabian Sea, and Eastern Boundary Upwelling Systems (e.g., Zonneveld et al., 2010; Jørgensen et al., 2022). Low pelagic oxygen concentrations will slow down the degradation of OM in the water column. Consequently, more reactive OM reaches the sediments and thus results in the occurrence of high sedimentary OM reactivity than predicted by the model in these regions.

References:

Zonneveld, K. A., Versteegh, G. J., Kasten, S., Eglinton, T. I., Emeis, K.-C., Huguet, C., Koch, B. P., de Lange, G. J., de Leeuw, J. W., and Middelburg, J. J.: Selective preservation of organic matter in marine environments; processes and impact on the sedimentary record, Biogeosciences, 7, 483-511, doi:10.5194/bg-7-483-2010, 2010.

Jørgensen, B. B., Wenzhöfer, F., Egger, M., and Glud, R. N.: Sediment oxygen consumption: Role in the global marine carbon cycle, Earth-science reviews, 228, 103987. doi:10.1016/j.earscirev.2022.103987, 2022.

**Reviewer 2#**

**Question 1:** When all is said and done, the main take home point of this work seems to be Fig. 3 and its subsequent discussion. Perhaps I'm being unduly harsh, but I'm not really sure I see a lot here that is really that new, as is indicated by the discussion toward the latter part of section 3.2. In some senses though, this consistency between the model results here and wide range of diverse observations regarding organic matter reactivity and composition is re-assuring, and in some ways this work does act to help "unify" these observations. On the other hand, in other places (lines 339 and 359), the authors note that "the l-RCM can be further used to calculate the budget of OM degradation at regional or global scales and assess the significance of the sedimentary carbon cycle on the hydrosphere and atmosphere." To me at least, adding such a calculation to this manuscript would be as (if not more) important and interesting as is Fig. 3. It could then be compared to other regional and global estimates of such quantities cited on lines 328-330, or reported more recently in Jørgensen et al. (2021, Earth-Sci. Rev. 228:103987). These estimates might also be a way of somewhat independently verifying how "good" this lognormal approach is, as compared to other models of sediment OM reactivity.

**Responses:** Thank you for your valuable comments. Currently, no robust quantitative framework exists that would allow predicting OM reactivity as a function of easily observable environmental parameters. The presented study's novelty lies in its unifying view and in contributing a new framework that allows predicting OM reactivity in data-poor areas based on readily available (or more easily obtainable) information. Such a framework currently needs to be improved and limits our abilities to constrain OM reactivity in global biogeochemical and/or Earth System Models. A global assessment of the benthic carbon budget in marine sediments would be a study in its own right. In the future, it is worth using our *l*-RCM to simulate the associated biogeochemical processes to better quantify the OM degradation and burial in marine sediments at regional or global scales.

**Question 2:** The overall manuscript is chopped up in such a way that makes it very hard for the reader (or at least me) to follow. Specifically, things discussed in the Supplementary Material section are not well-referenced in the text, and I was very confused when I first started reading the main text, until I realized I had better go through the Supplementary Material section first. In revising the manuscript, I would work to restructure the work as a whole so that it flows better, i.e., better link the main text and the Supplementary Materials sections and also minimize repetition in places.

**Question 3:** The math in the supplementary section is very dense and confusing in places (also see point 5 below).

**Responses (2 and 3):** Thank you for raising this insight question. We have reorganized the content of the main text and the supplementary material, especially rewritten the methods section. In the previous manuscript version, the method description was missing some key information. We therefore added several additional figures and schematic diagrams. For example: 1. we added the distribution map of our simulated sites to the main text (Fig.2); 2. we also moved the fitting result of global sedimentation rate to the main text (Fig.3).

**Question 4:** Maybe I'm missing something, but there seem to be two definitions of (eqn. 4 and eqn. 7, which is the same as eqn. S3) and in plots like Figs. 2 and 3 it's not clear which is being used. This confusion needs to be cleared up in the revisions.

**Responses:** Thank you for the suggestion. We have reorganized the content of the main text and the supplementary material, especially rewritten the methods section. The eqn. 4, eqn. 7, and eqn. S3 in the previous manuscript were used to calculated regional OM reactivity, which was complexity. In the latest manuscript, we have used a relatively concise method to calculate OM reactivity in the different regions (see the response to **Question 15**).

**Question 5:** The quantity $\rho(\ln(k))$ or $\rho(k)$ is plotted in several places (Figs.1C, S4, S5, S12, S13). This parameter is not clearly defined in the text (maybe I missed it), and it

is also not clear how it relates to other parameters being looked at here (this comment may actually be a specific example of the general concern noted in point 3 above).

**Responses:** We have modified the text and now consistently refer to $(\rho(k))$ in the whole main text and supplementary material. In addition, we also added a clear definition of $\rho(k)$ (probability density function of reactivity $k$, Fig. 1 and **Line 133**) in the Section of Introduction.

**Question 6:** The referencing in the early part of the manuscript needs to be cleaned up. You don't write "… Washington and Jefferson (Washington and Jefferson, 1776) said …" but rather "… Washington and Jefferson (1776) said …".   Also references with 2 authors do not use et al. (e.g., see lines 138 and 179), and again remove the author names from inside the parenthetical statement.

**Responses:** We have checked and revised it accordingly.

**Question 7:** (82) – I'm not sure where this R2 comes from.

**Responses:** We have added Table 1 and Fig.5 to show the fitting results of the $\gamma$-RCM and the $\gamma$-RCM.

**Table 1. List of model parameters and coefficients of determination ($R^2$) for the fitting result of $\gamma$-RCM and $l$-RCM.**

| Core | $\gamma$-RCM | | | $l$-RCM | | |
|---|---|---|---|---|---|---|
| | $v$ (-) | $a$ (yr) | $R^2$ | $\mu$ (yr$^{-1}$) | $\sigma$ (-) | $R^2$ |
| Foam | 0.152 | 4.2 | 0.930 | $2.2\times10^{-3}$ | 3.725 | 0.923 |
| SCR-44 | 0.202 | 70.4 | 0.929 | $4.4\times10^{-4}$ | 2.706 | 0.922 |
| BX-6 | 0.278 | 22.5 | 0.929 | $2.24\times10^{-3}$ | 2.031 | 0.936 |
| PC2&TW2 | 0.052 | 0.16 | 0.937 | $5.5\times10^{-5}$ | 6.688 | 0.947 |
| 10141&2 | 0.193 | 10184 | 0.935 | $1.9\times10^{-6}$ | 3.289 | 0.936 |
| 7706-41K | 0.910 | 141.3 | 0.974 | $9.5\times10^{-3}$ | 0.899 | 0.972 |
| 7706-36 | 0.804 | 231.7 | 0.978 | $4.79\times10^{-4}$ | 1.089 | 0.980 |
| DSDP58 | 1.080 | 20224 | 0.917 | $6.11\times10^{-5}$ | 1.663 | 0.921 |

[Figure]

**Figure 5. Fitting results of the *l*-RCM and the *γ*-RCM.** The pink dots are the measured OM data, the red lines are *l*-RCM fitting results, and the blue lines are *γ*-RCM fitting results.

**Question 8:** (127-9) – I think I know what is meant here, but it could be worded better, and a reference or two might be useful. Also you can see the tail that is referred to here in Fig. S13 (vs. Fig. S4) – perhaps this point could somehow be included here?

**Responses:** Thank you for raising this insight question. We have reorganized the content of the main text and the supplementary material, especially rewritten the methods section.

**Question 9:** (128) – I think is better referred to as the mean rate constant for bulk OM degradation.

**Responses:** We have checked and revised it. Please see this change in **Line 150**.

**Question 10:** (145) – Does $\omega$ vary with depth at any of these sites, and if so, is this a problem?

**Responses:** Thank you for your valuable comments. We agree that the sediment compaction impacts burial rates and thus OM degradation. The compaction process leads to the change of porosity, which in turn affects the burial time of OM. In Lines 159-169, we have added the effect of porosity on burial time during OM degradation.

**Question 11:** (160) – I'm having a hard time understanding how F-i(k,0) is defined, both here and in sections S4 and S5. For example, here it seems like the i subscript in eqn. (6) refers to each 1°x1° grid cell and that this is then used to calculate the values for each grid cell plotted in Fig. 3. On the other hand, eqn. (S2) is almost identical to eqn. (6) but this equation refers to this (line 99, SM) as the "distribution of OM reactivity at the regional to global scale". What am I missing here?

**Responses:** Thank you for pointing this out. We have reorganized the content of the main text and the supplementary material. Due to the complexity and unclear expression of the calculation method used in the previous manuscript, we have used a relatively concise method to calculate OM reactivity in the different regions (see the response to **Question 15**).

**Question 12:** (179) – The 8 data sets plotted in Fig. S3 do not come from the Westrich and Berner (1984) paper.

**Responses:** Thank you for pointing out this issue. We checked the source of the data, and they were reported in Boudreau and Ruddick. (1991) (**Line 209**).

**Question 13:** (292-299) – Separating out certain regions (e.g., EWEP, SWAF, NWAM, and ARBS) in Fig. 2B based on the discussion here of R2 values from different modeling efforts seems a bit suspect. It might also be interpreted as applying a "2-G" approach to the l-RCM (i.e., different types of OM produced in different parts of the oceans show different trends in reactivity). In the end though if all of the data in Fig. 2B were fit to a single straight line and then used to recalculate Fig. 3 I wonder if the results would be that much different. I don't want to make a big deal about this, but this is something to consider.

**Responses:** Yes, we fully agree. We initially tried to describe the global distribution of OM reactivity with a regression curve, as shown in the figure below. However, the resulting $<k>$-$w$ function captures the weak general trend of decreasing OM reactivity

with decreasing water depth but fails to predict the observed regional patterns in OM reactivity (e.g., especially in the deeper EWEP, SWAF, NWAM, and ARBS).

[Figure]

**Figure (1) Log-log plot of $\omega$ and $<k>$ with a single straight line; (2) Distribution of $<k>$ in the global ocean**

The reactivity of OM is influenced by several environmental factors, and a single relationship between reactivity and sedimentation rate alone is not a good proxy for the complex environmental controls on OM reactivity. Therefore, we tried two regression curves to map and better capture the global OM reactivity distribution.

**Question 14:** (297) – The phase "is less quality" needs revision.

**Responses:** Thank you for pointing this out. We have replaced it accordingly (**Lines 348-352**). "**Despite the poor relationship** between $<k>$ and $\omega$ for special regions".

**Question 15:** (109-110) – How exactly are the curves shown in Fig. S12 obtained? Is each curve used in each of the 30 regions to generate the distribution of values shown in Fig. 3? If so, does this then mean that the observed variation in within each region is driven solely by differences in water depth and sedimentation rate? Please clarify this in the revisions.

**Question 16:** (139, SM) – What is an "irregular distribution"?

**Question 17:** (142-148) – I think the authors are simply saying here that eqn. (S4) is a trapezoidal approximation used to numerically integrate eqn. (S3). If so, I would say it as such (here and in the main text near line 172). As written, it sounds odd to talk about dividing the OM into 1000 fractions (which isn't really being done), especially after reading the Introduction where the authors talk about problems with multi-G models.

I realize the components of a multi-G model are not conceptually the same as the components (or fractions) being discussed here in this calculation. At the same time, since there is no need to use terminology that even hints at these similarities, I would modify this text to avoid any unnecessary confusion.

**Responses (15-17):** Thanks to the reviewer for helpful comments. Due to the complexity and unclear expression of the calculation method used in the early part of the manuscript, we have reorganized the structure of the manuscript and used a relatively concise method to calculate OM reactivity in the different regions ($<K_{regions}>$).

In the revised manuscript, $\omega$ was estimated based on the empirical relationship between $\omega$ ($\omega$ in cm yr$^{-1}$) and the water depth ($z$ in m) (Eq.7, Fig.3), and the values of $<K_{region}>$ in the 30 regions (Table 2) were calculated according to the relationship between water depth, $\omega$, and $<k>$ (Fig.3 and Fig.8C). Please see the detailed modification in **Lines 177-201** and the Section of 3.3 "Global distribution patterns of OM reactivity".

[Figure]

**Figure 3. Relationship between Sedimentation rate ($w$) and water depth ($z$ in m).**

[Figure]

**Figure 8. Global distribution patterns of OM reactivity.**

---

## Author Response (AR2)

**Responses to the Editor**

Dear Prof. Jack Middelburg,

Firstly, we wish to express our gratitude to you for your suggestions and comments that helped us to improve our manuscript. We have addressed all comments point-by-point as outlined in detail below.

Sincerely,

Zijun Wu

**Question 1:** L.44, benthic carbon burial over geological timescales.

**Response:** Corrected as suggested. Please see this change in **Lines 44**.

**Question 2:** L. 66, replace on the other hand with however.

**Response:** Corrected (**Line 66**).

**Question 3:** L. 152, is this difference between median and mean an approximate correction or absolute? I believe the former.

**Response:** We have checked and revised this sentence (**Lines 150-151**).

**Question 4:** L. 178-180: "The correction factor for skewness bias" sentence needs a citation and more explanation. Is too cryptic in its present form.

**Response:** We have reorganized this sentence in the revised manuscript, including adding relevant references and Equation 7 to describe the calculation of the correction factor (**Lines 179-183**).

$$f_c = e^{2.65 \times s^2} \tag{Eq.7}$$

**Question 5:** L. 386: identifying

**Response:** Corrected (**Line 388**).

**Question 6:** L. 417-419: rewrite sentence. You start to refer to two relationship and only mention one. I guess that you mean k with w and w with z.

**Response:** Thank you for pointing this out. We have rewritten this sentence as "… two empirical relationships of $<k>$ with $\omega$ and $\omega$ with $z$" (**Lines 419-421**).